# Concurrent and Concordant Anal and Oral Human PapillomaVirus Infections Are Not Associated with Sexual Behavior in At-Risk Males

**DOI:** 10.3390/pathogens10101254

**Published:** 2021-09-28

**Authors:** Francesca Rollo, Alessandra Latini, Maria Benevolo, Amalia Giglio, Eugenia Giuliani, Barbara Pichi, Raul Pellini, Massimo Giuliani, Maria Gabriella Donà

**Affiliations:** 1Pathology Department, Regina Elena National Cancer Institute IRCCS, Via Elio Chianesi 53, 00144 Rome, Italy; francesca.rollo@ifo.gov.it (F.R.); maria.benevolo@ifo.gov.it (M.B.); 2STI/HIV Unit, San Gallicano Dermatological Institute IRCCS, Via Elio Chianesi 53, 00144 Rome, Italy; alessandra.latini@ifo.gov.it (A.L.); mariagabriella.dona@ifo.gov.it (M.G.D.); 3Microbiology and Pathology Department, San Gallicano Dermatological Institute IRCCS, Via Elio Chianesi 53, 00144 Rome, Italy; amalia.giglio@ifo.gov.it; 4Scientific Direction, San Gallicano Dermatological Institute IRCCS, Via Elio Chianesi 53, 00144 Rome, Italy; eugenia.giuliani@ifo.gov.it; 5Otolaryngology Head Neck Surgery Department, Regina Elena National Cancer Institute IRCCS, Via Elio Chianesi 53, 00144 Rome, Italy; barbara.pichi@ifo.gov.it (B.P.); raul.pellini@ifo.gov.it (R.P.)

**Keywords:** HPV, concordance, concurrence, MSM, HIV, anal, oral

## Abstract

Men who have sex with men (MSM) harbor the highest prevalence of anal and oral Human Papillomavirus (HPV) infection, particularly if HIV-infected. We investigated anal and oral HPV infections in HIV-infected and HIV-uninfected MSM, to assess concurrent (HPV detected at both sites, irrespective of the genotypes), and concordant infections (same genotype[s] detected at both sites). Matched anal and oral samples from 161 MSM (85 HIV-infected, and 76 HIV-uninfected) were tested with the Linear Array. Determinants of concurrent and concordant infections were evaluated using logistic regression. Anal infections were 4 to 7 times more frequent than oral infections in both study groups (*p* < 0.0001). Concurrent infections were not significantly different in HIV-infected (25.9%) and HIV-uninfected MSM (17.1%), *p* = 0.18. A concordant infection was found in 15 MSM (9.3%). Concordance was for one genotype in 14 individuals and for four genotypes in the remaining subject. In the overall population, only age was independently associated with a concurrent infection (AOR = 3.10, 95% CI: 1.34–7.19 for >39 vs. ≤39 years). None of the parameters of sexual behavior showed independent association with concordant infections. Among MSM, concordant anal and oral HPV infections do not seem to be explained by sexual behavior, but might derive from sequential acquisition by autoinoculation.

## 1. Introduction

Persistent infections by mucosal high-risk Human Papillomaviruses (HPVs) are known to cause cancer at several anatomic sites [1]. Current research in the HPV field is largely focusing on anal and oral infections because of their causal association with anal and head and neck carcinomas, respectively.

Anal and oral HPV infections have been extensively investigated in different sexually active populations [2,3]. The epidemiology of these two infections, separately, has been largely investigated, evidencing the role of sexual exposure in the acquisition and transmission dynamics, particularly in groups with high frequency of sexual contacts, partners, and practices, such as men who have sex with men (MSM) [2,4]. They in fact harbor the highest prevalence of anal and oral infection [3,5]. Unfortunately, little is known about the correlation between these two infections in MSM. Several studies have explored co-occurrence, type-specific concordance, and sequential acquisition of multisite HPV infections (oral and genital, anal and genital) among women and men who have sex with women, suggesting that infections at these sites may not be independent [6,7,8,9,10,11,12]. Although they may result from sexual behavior, they may also be consequent to autoinoculation from one anatomical site to the other [7,10,13].

We aimed to investigate whether or not HPV infections simultaneously occur at anal and oral sites in sexually active MSM, who habitually practice both anal and oral sex. This may contribute to clarifying to what extent concurrent and concordant anal and oral HPV infections can be found among MSM. In addition, we evaluated the determinants of anal and oral HPV co-occurrence, also exploring possible associations with sexual behavior. This may explain whether or not sexual transmission plays a role in the occurrence of simultaneous anal and oral infections.

## 2. Results

### 2.1. Study Population

During the study period, matched samples for HPV DNA detection at the anal and oral level were available from 85 (52.8%) HIV-infected and 76 (47.2%) HIV-uninfected MSM, for a total of 161 enrolled subjects and 322 paired samples.

The median of nadir CD4+ T-cell count for the HIV-infected participants was 300 cells/mm^3^ (interquartile range [IQR]: 201–394). At enrollment, almost all these individuals were on combined antiretroviral therapy (cART) (*n* = 81, 95.3%) and their median CD4+ T-cell count was 622 cells/mm^3^ (IQR: 474–813). In addition, 75/81 (92.6%) had an undetectable (<40 copies/mL) HIV-RNA viral load at the moment of samples collection.

Demographic and behavioral characteristics of the MSM stratified by the HIV status are presented in Table 1.

The median age of the HIV-infected individuals was 44 (IQR: 34–49), as compared with 37 years (IQR: 32–45) for the HIV-uninfected individuals (*p* = 0.007). The HIV-infected and HIV-uninfected MSM referred a similar age at sexual debut, having had their first intercourse with a man at 18 (IQR: 17–20) and 20 years (IQR: 17–24), respectively (*p* = 0.16).

Among HIV-infected MSM, the median number of lifetime partners for any sex did not significantly differ from that of the HIV-uninfected individuals (*p* = 0.40), nor did the median number of lifetime receptive oral sex partners (*p* = 0.20). Differently, HIV-infected individuals had a significantly lower number of recent any sex partners (*p* = 0.0002) and receptive oral sex partners (*p* = 0.01), compared with the HIV-uninfected counterparts. All the HIV-uninfected MSM and 97.6% of the HIV-infected MSM reported to practice receptive oral sex, but the HIV-uninfected individuals had practiced it more frequently in the previous 6 months (*p* = 0.006). The condom use in recent oral sex did not significantly differ between the study groups. In fact, 95.7% and 97.3% of the HIV-infected and HIV-uninfected MSM, respectively, practiced condomless oral sex. The majority of the participants referred to practice receptive anal sex (78.8% and 85.5%, respectively, *p* = 0.27), and the use of condom during this type of intercourse did not significantly differ between the two groups (*p* = 0.83), with less than half of the subjects reporting no condom use.

### 2.2. Prevalence of Anal and Oral HPV Infection in HIV-Infected and HIV-Uninfected MSM

All the oral rinses and the anal samples gave a valid HPV test result. Prevalence rates of oral and anal infections according to the HIV status are indicated in Figure 1.

At a glance, the prevalence of HPV infection was significantly higher in the anal than oral site both among HIV-infected and HIV-uninfected individuals, regardless of the HPV group analyzed. Among HIV-infected MSM, 80/85 (94.1%, 95% CI: 87.0–97.5) and 22/85 (25.9%, 95% CI:17.8–36.1) individuals were positive for any HPV at the anal and oral site, respectively (*p* < 0.0001). High-risk HPVs were detected in 68/85 (80.0%, 95%CI:70.3–87.1) of anal and 10/85 (11.8%, 95% CI: 6.5–20.3) of oral samples (*p* < 0.0001), while 32/85 (37.6%, 95% CI: 28.1–48.3) and 2/85 (2.4%, 95% CI: 0.6–8.2) MSM were HPV16-positive at the anal and oral site, respectively (*p* < 0.0001). Of the HIV-uninfected counterparts, 60/76 (78.9%, 95% CI: 68.5–86.6) and 15/76 (19.7%, 95% CI: 12.3–30.0) were HPV-positive at anal and oral sites, respectively (*p* < 0.0001). Forty-eight individuals (63.1%, 95%CI:51.9–73.1) were positive for high-risk HPVs at anal level and the respective figure for the oral site was 9 (11.8%, 95% CI: 6.4–21.0) (*p* < 0.0001). Regarding HPV16, less than a quarter (22.4%, 95% CI: 14.5–32.9) harbored the anal infection, while 7/76 (9.2%, 95% CI: 4.5–17.8) harbored the oral one.

Among the HIV-infected MSM, 10 HPV genotypes were only detected in anal samples (among these were high-risk HPVs 31, 35, 52, 56, and 58), whereas in none of the cases was an HPV genotype exclusively found in oral samples. Similarly, 20 HPV genotypes were only detected in anal samples from HIV-uninfected MSM (among these were high-risk HPVs 18, 31, 33, 39, 52, 59, and 66), whereas no HPV genotype was exclusively found in oral samples.

### 2.3. Concurrent and Concordant Anal–Oral HPV Infections

HPV infections at both sites were investigated at patient level to assess the prevalence of concurrent and concordant infections. The majority of the participants (68.2% of the HIV-infected and 64.5% of the HIV-uninfected MSM) harbored nonconcurrent infections, i.e., they were HPV-positive exclusively at anal or oral level (Figure 2). However, the latter case was infrequent, being observed only in two out of 76 HIV-uninfected subjects (2.6%). The remaining participants of both study groups were either negative or positive at both sites.

Overall, 35 out of 161 MSM (21.7%) harbored a concurrent infection (Table 2). In particular, this was detected among 22/85 HIV-infected (25.9%, 95% CI: 17.8–36.1) and 13/76 HIV-uninfected MSM (17.1%, 95% CI: 10.3–27.1), respectively, with no significant difference between the two study groups (*p* = 0.18).

Similarly, concordant infections were more frequent among HIV-infected subjects, being detected in 11.8% (95% CI: 6.5–20.3) of them vs. 6.6% (95% CI: 2.8–14.5) of the HIV-uninfected counterparts, without such difference reaching statistical significance (*p* = 0.26). Among the 15 MSM with a concordant infection, 14 were concordant for one genotype: HPV16 was simultaneously detected in both the anal and oral sample in 4 individuals; HPV18 and HPV84 in 2 individuals each; HPV45, 59, 62, 70, 72, and 73 in 1 individual each. The remaining subject, who was HIV-infected, was concordant for 4 genotypes (HPVs 11, 16, 18, and 81).

### 2.4. Determinants of Concurrent and Concordant Anal–Oral HPV Infections

The results of univariate and multivariate analyses for the concurrent and concordant anal–oral HPV infections are shown for the overall population in Table 3.

In univariate analysis, age (COR: 3.33, 95% CI: 1.48–7.52, for those over 39 vs. ≤39 years), number of lifetime partners for any sex (COR: 2.32, 95% CI: 1.06–4.07, for those reporting more than 75 vs. ≤75 partners), and for receptive oral sex (COR: 2.91, 95% CI: 1.33–6.38, for those having more than 50 vs. ≤50 partners) were significantly associated with a concurrent infection. In multivariate analysis, age was the only variable that remained independently associated with the outcome. In detail, those over 39 years of age showed approximately 3-fold higher odds of having a concurrent infection compared with younger individuals (AOR: 3.10, 95% CI: 1.34–7.19).

Despite the fact that in univariate analysis, those with a higher number of lifetime partners for any sex (COR: 4.59, 95% CI: 1.24–16.94, for those reporting more than 75 vs. ≤75 partners) and for receptive oral sex (COR: 3.73, 95% CI: 1.13–12.25, for those having more than 50 vs. ≤50 partners) showed increased odds of concordant infection, in the multivariate model these associations did not reach statistical significance.

## 3. Discussion

Here, we investigated oral and anal HPV infections in HIV-infected and HIV-uninfected MSM subjected to simultaneous sampling at the two anatomical sites. We also assessed the predictors associated with concurrent and concordant infections.

Previously, we investigated anal and oral HPV infections in MSM populations, and compared HIV-infected vs. HIV-uninfected subjects in terms of prevalence at the two sites [14,15]. Therefore, the comparison between HIV-infected and HIV-uninfected MSM will not be further discussed here.

In the present study, we found that prevalence of i) any type of HPV, ii) high-risk types and iii) HPV16 was significantly higher at anal than oral level, both among HIV-infected and HIV-uninfected MSM. In fact, anal HPV was found to exceed oral HPV prevalence by 4 to 7 times. This is consistent with findings by others who compared anal vs. oral infection in MSM [16,17,18,19,20]. It is interesting to note that the substantial disproportion between anal and oral prevalence found in this study cannot be justified by the sexual behavior of the participants. In fact, receptive anal and receptive oral sex were practiced by very similar proportions of subjects. In addition, the fraction of MSM who referred to have condomless intercourses was lower for anal sex (around 40%) than oral sex (over 95%). Our and other similar findings might be due to a combination of the diverse risk of HPV acquisition, along with a different natural history of the infection at the two anatomical sites [21]. Indeed, in different yet similar MSM populations, we have previously observed a higher incidence of anal HPV infection and, vice versa, a higher clearance of oral HPV infection [22,23]. Notably, the same results have been obtained in a study that simultaneously evaluated anal and oral HPV natural history in the same population of HIV-infected individuals [24]. One of the factors contributing to the higher incidence of anal infection may be represented by the fact that microtraumas occurring during anal sex could promote virus access to the squamous epithelium basal layer, where HPV target cells reside. As demonstrated by the abovementioned study by Beachler et al. [24], anal HPV is also more likely to persist compared to the oral infection. The mucosal immune response and the continuous salivary flow in the oral cavity may be responsible for the faster clearance of HPV infection at this anatomic district [5].

The majority of our participants (approximately 65%) harbored nonconcurrent infections. In all the HIV-infected MSM with nonconcurrent infections, HPV was exclusively detected in the anal sample. Differently, 2.6% of the HIV-uninfected MSM harbored HPV only at oral level.

A concurrent infection was found in one-fifth of all the subjects. HIV-infected and HIV-uninfected MSM did not differ significantly in the prevalence of concurrent infections. In other studies conducted on MSM, concurrent infections were either less [16,19] or more common [20] than in ours, but divergent results may find several explanations. Methodological aspects (e.g., sampling procedure, HPV detection method, and HPVs included in the analysis) and characteristics of the study population may all affect the estimates.

Concordant infections in our study subjects were infrequent, being found in about 9% of all MSM. Nevertheless, they were twice more common than in the investigation by Steinau et al. who also analyzed samples from HIV-positive and -negative MSM using our same HPV assay [19]. In line with this U.S. study, however, we observed that concordant infections were more frequent in HIV-infected MSM, although not significantly. Other investigations that compared anogenital vs. oral instead of anal vs. oral infections in MSM found no concordance [2,25], although Kahn et al. only analyzed HPV types included in the nonavalent vaccine. Nonetheless, these findings suggest that among MSM, concordance is more likely between anal and oral than anogenital and oral sites.

The fact that concordant infections, despite being infrequent, were still observed in our population may find several explanations. MSM could acquire the same HPV genotype(s) at both anatomical sites by practicing both receptive anal and oral sex with the same partner, who thus transmits his genital infection. However, none of the parameters of sexual behavior investigated in this study showed a significant association with concordant infections. In addition, having a concordant infection does not seem to be explained by the type of recent partnership, given that, of the 15 MSM harboring concordant infections, three had exclusively one stable partner, six had exclusively occasional partners, and three had both. Sequential HPV acquisition at both anal and oral level through nonsexual routes, e.g., autoinoculation, is thus likely, as suggested also for oral and genital HPV infection in men [10]. In agreement with this, a very recent study on orogenital concordance for high-risk HPVs in men found no predictors of concordance among sexual behavioral variables [9]. Finally, there is also the possibility that the same HPV(s) were detected at both sites by chance, although this can hardly explain concordance for four genotypes as observed in one participant.

Interestingly, the only independent association with a concurrent infection was observed with age. We observed that older age (≥39 years) tripled the odds of having a concurrent infection. Despite the fact that the predictors were analyzed on the overall study population, this increase by age is consistent with our previous findings on oral HPV infection that revealed an increased risk of acquiring any HPV by older HIV-uninfected MSM and a reduced clearance rate displayed by older HIV-infected MSM [23]. In addition, a higher prevalence of HPV in the anal canal of older MSM has been also observed [26,27]. Taken together, these features may partially explain the higher probability of simultaneously detecting HPV at anal and oral level in older MSM.

Limitations of this study include the restricted number of participants, which did not allow us to perform a predictor analysis of concurrent and concordant infections stratifying by HIV status. Moreover, our population reported multiple partnerships, which makes it difficult to disentagle the contribution of sexual behavior on the simultaneous positivity at both sites. However, the reliability of our findings is supported by the use of the same very sensitive HPV genotyping test both for anal and oral samples. In addition, the availability of information regarding several parameters of sexual behavior allowed us to explore whether they were determinants of concurrent and concordant infections, instead of merely providing prevalence data.

In conclusion, analyzing paired anal and oral samples of MSM, we observed that the prevalence of anal HPV significantly exceeds prevalence of oral HPV, irrespective of the HIV status of the participants. Concurrent infections at anal and oral sites were found in around one-fifth of the study subjects. Age was the only predictor of concurrent infections. Genotype-specific concordance was infrequent but still observed. Intriguingly, it did not show independent association with any of the investigated covariates of sexual behavior. These findings do not support the role of sexual behavior as a determinant of concordant anal and oral HPV infections in MSM, although this cannot be entirely ruled out. Sequential acquisition of the same HPV genotype(s) at two different anatomical sites might occur through nonsexual routes, such as autoinoculation from one site to the other. Longitudinal studies simultaneously investigating the infection at both sites are pivotal to unravel the tangle.

## 4. Materials and Methods

### 4.1. Study Population

The study population included HIV-infected and HIV-uninfected MSM (at least one sexual male partner in the previous 6 months) aged ≥18 years, who attended the STI/HIV Unit of the San Gallicano Dermatological Institute in Rome (Italy) and had been recruited in two prospective studies: the Surveillance Program of Anal Intraepithelial Neoplasia (SAIN) and the Oral/Oropharyngeal HPV in Men at Risk (OHMAR). Inclusion criteria for the abovementioned studies have been detailed elsewhere [28,29,30]. Inclusion criteria for the present study were as follows: (i) having both an anal and oral sample collected on the same day; (ii) having a valid HPV test result on both samples. Information about sociodemographic profiles, sexual behavior (lifetime and recent, i.e., regarding the previous 6 months), and clinical history were obtained by face-to-face interviews. Clinical data about the HIV-infected subjects were retrieved from the medical records. Written informed consent was obtained from all MSM enrolled.

### 4.2. Clinical Sample Collection

The anal samples were collected using a sterile Dacron swab, which was introduced and rotated in the anal canal for at least 60 s. The swab was subsequently shaken into the PreservCyt cytology medium (Hologic, Pomezia, Italy) in order to displace the epithelial cells. Oral samples were collected by asking participants to rinse and gargle for 30 s with 15 mL of Listerine mouthwash, finally expectorating in a 50 mL Falcon tube. Once obtained after centrifugation, the pellet was washed twice with PreservCyt (Hologic, Pomezia, Italy) and finally resuspended in 2 mL of PreservCyt.

### 4.3. HPV-DNA Detection and Genotyping

The Linear Array HPV genotyping test (Roche Diagnostics, Milan, Italy) was used to test both oral and anal samples. This assay detects the DNA of 37 different mucosal HPVs, including those defined as “high-risk” by IARC (i.e., 16, 18, 31, 33, 35, 39, 45, 51, 52, 56, 58, and 59). In accordance with the datasheet instructions, the AmpliLute Liquid Media Extraction kit (Roche Diagnostics, Milan, Italy) was used to obtain the total nucleic acids from each anal and oral sample stored in PreservCyt. Then, 50 μL of each extract was used as a template for amplification. The hybridization and detection were performed using an automated processor (ProfiBlot T48, Tecan, Männedorf, Switzerland). As per manufacturer instructions, samples were considered as HPV-positive if at least one HPV hybridization band was detected, irrespective of the presence of the beta-globin controls.

### 4.4. Data Analysis

Summary statistics (median, IQR) were used to describe the study groups. Median values were compared by the Mann–Whitney test. The distributions of the categorical variables in the two study groups were compared by chi-square test.

For the purposes of HPV infection analysis, anal and oral infections were analyzed separately and for each study group. For both types of samples, individuals were considered positive for any HPV when harboring at least one type among the 37 detectable by the Linear Array. The classification of the high-risk group was as follows: HPVs 16, 18, 31, 33, 35, 39, 45, 51, 52, 56, 58, 59, 66, and 68. Prevalences of anal vs. oral infection within each study group were compared by chi-square test.

HPV infection was defined: (i) concurrent when the matched anal and oral samples were both HPV-positive, irrespective of the genotype(s); (ii) concordant when the matched anal and oral samples had at least one genotype in common. Univariate and multivariate analysis were conducted on the overall study population, using HIV status as a covariate. Predictors of concurrent and concordant infections were assessed separately. For the univariate and multivariate analysis, continuous variables (age, age at first sex, number of lifetime and recent partners for any and oral sex) were recoded as categorical variables. The median value for the overall study population was used as a cutoff to obtain two categories, i.e., (i) more than and (ii) less than or equal to the median value. Alcohol consumption was classified as previously detailed [23]. Crude odds ratios (COR) and a 95% confidence interval (CI) were calculated. The multivariable model included all the covariates that showed a *p* < 0.05 at univariate analysis. Adjusted OR (AOR) and corresponding 95% CI were calculated. The statistical analyses were conducted using MedCalc Statistical Software version 20.009 (MedCalc Software Ltd., Ostend, Belgium; https://www.medcalc.org; accessed on 15 June 2021).

## Figures and Tables

**Figure 1 pathogens-10-01254-f001:**
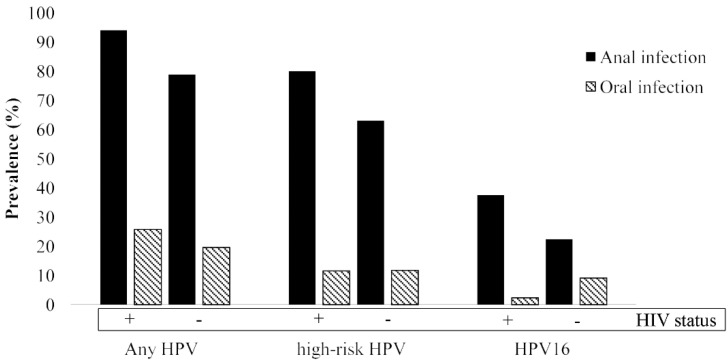
Prevalence of anal and oral HPV infection by any HPV, high-risk HPVs, and HPV16 among HIV-infected and HIV-uninfected MSM.

**Figure 2 pathogens-10-01254-f002:**
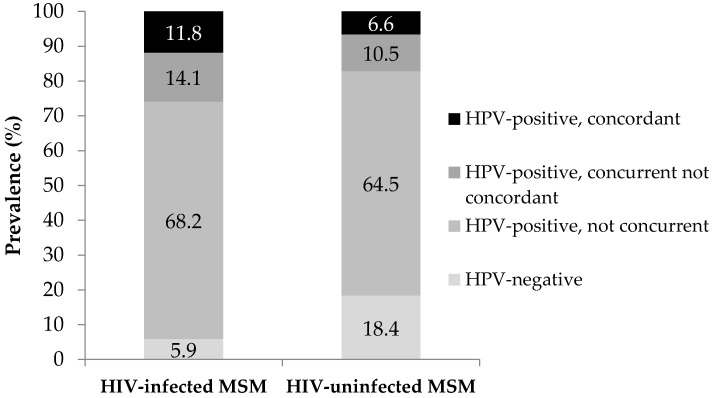
Proportions of MSM who were found to be negative both in the anal and oral sample (HPV-negative), positive either in the anal or oral sample (HPV-positive, nonconcurrent) and positive both in the anal and oral sample [(i) HPV-positive, concurrent not concordant and (ii) HPV-positive, concordant]. Data are shown for the two study groups, separately.

**Table 1 pathogens-10-01254-t001:** Sociodemographic and behavioral characteristics of the study population according to HIV status.

	HIV-Infected MSM		HIV-Uninfected MSM		
	***n* = 85**		***n* = 76**		
	**median (IQR)**				** *p* ** **value**
**Age, years**	44 (34–49)		37 (32–45)		**0.007**
**Age at first sex, years**	18 (17–20)		20 (17–24)		0.16
**N. lifetime partners for any sex**	100 (34–300)		70 (33–120)		0.40
**N. lifetime partners for receptive oral sex**	60 (15–150)		40 (15–86)		0.20
**N. recent partners for any sex** ^1^	2 (1–8)		6 (3–13)		**0.0002**
**N. recent partners for receptive oral sex** ^1^	2 (1–6)		4 (1–10)		**0.01**
	** *n* **	**%**	** *n* **	**%**	** *p* ** **value**
**Caucasian ethnicity**	82	96.5	74	97.4	0.52
**Graduate education**	31	36.5	40	52.6	**0.039**
**Annual income**					0.17
>12,000 euros	14	16.5	7	9.2	
Up to 12,000 euros	71	83.5	69	90.8	
**Alcohol consumption**					0.13
Light/non drinker	61	71.8	46	60.5	
Moderate and heavy	24	28.2	30	39.5	
**Smoking status**					0.33
Current	40	47.1	27	35.5	
Former	8	9.4	8	10.5	
Never	37	43.5	41	53.9	
**Receptive oral sex (ever)**	83	97.6	76	100.0	0.18
**Recent receptive oral sex**	70	82.4	73	96.1	**0.006**
**Condomless recent oral sex** ^2^	67	95.7	71	97.3	0.62
**Recent receptive anal sex**	67	78.8	65	85.5	0.27
**Condomless receptive anal sex** ^3^	29	43.3	27	41.5	0.83
**History of STI**					**0.01**
No	15	17.6	29	38.2	
Ano-genital warts	32	37.6	20	26.3	
Other than ano-genital warts	38	44.7	27	35.5	

^1^ during the previous 6 months; ^2^ percentages calculated over the number of those reporting recent receptive oral sex (*n* = 70 and *n* = 73 for HIV-infected and HIV-uninfected, respectively); ^3^ percentages calculated over the number of those reporting recent receptive anal sex (*n* = 67 and *n* = 65 for HIV-infected and HIV-uninfected, respectively); statistically significant differences are shown in bold.

**Table 2 pathogens-10-01254-t002:** Concurrent and concordant anal and oral HPV infections among 161 MSM, according to HIV status.

	Anal–Oral HPV Infection			
MSM	Concurrent ^1^ *n*/N (%)	*p* Value ^2^	Concordant ^3^ *n*/N (%)	*p* Value ^2^
**HIV-infected**	22/85 (25.9)	0.18	10/85 (11.8)	0.26
**HIV-uninfected**	13/76 (17.1)		5/76 (6.6)	
**Total**	35/161 (21.7)		15/161 (9.3)	

^1^ Concurrent: simultaneous detection of at least one HPV in both samples; ^2^
*p* value for HIV-infected vs. HIV-uninfected subjects; ^3^ Concordant: simultaneous detection of the same HPV type(s) in both samples.

**Table 3 pathogens-10-01254-t003:** Univariate and multivariate analyses for the association of the selected covariates with concurrent and concordant anal–oral HPV infections.

Concurrent Anal–Oral HPV Infection						
Variable	COR	95% CI	*p* Value	AOR	95% CI	*p* Value
**Age, >39 vs. ≤39 years**	**3.33**	**1.48–7.52**	**0.004**	**3.10**	**1.34–7.19**	**0.008**
**Alcohol consumption, moderate/heavy vs. light/no**	1.22	0.56–2.67	0.61			
**Smoking, current vs. never/former**	1.44	0.68–3.05	0.35			
**Age at first sex, ≤19 vs. >19 years**	1.51	0.71–3.19	0.28			
**N. lifetime partners, any sex (>75 vs. ≤75)**	**2.32**	**1.06–4.07**	**0.03**	0.90	0.26–3.13	0.87
**N. recent partners, any sex (>4 vs. ≤4)**	1.09	0.52–2.31	0.82			
**N. lifetime partners, receptive oral sex (>50 vs. ≤50)**	**2.91**	**1.33–6.38**	**0.008**	2.87	0.84–9.77	0.09
**N. recent partners, receptive oral sex (>3 vs. ≤3)**	1.09	0.51–2.31	0.83			
**Receptive anal sex, yes vs. no**	1.41	0.50–4.02	0.52			
**Condomless receptive anal sex, yes vs. no**	1.85	0.81–4.20	0.14			
**Receptive oral sex (ever), yes vs. no**	1.43	0.07–30.38	0.82			
**Receptive oral sex (recent), yes vs. no**	0.97	0.30–3.15	0.96			
**Condomless receptive oral sex, yes vs. no**	0.40	0.06–2.50	0.33			
**Ano-genital wart history, yes vs. no**	1.32	0.60–2.88	0.49			
**HIV status, positive vs. negative**	0.59	0.27–1.28	0.18			
**Age, >39 vs. ≤39 years**	2.23	0.73–6.85	0.16			
**Alcohol consumption, moderate/heavy vs. light/no**	1.00	0.32–3.06	0.98			
**Smoking, current vs. never/former**	2.28	0.77–6-74	0.13			
**Age at first sex, ≤19 vs. >19 years**	0.44	0.13–1.45	0.18			
**N. lifetime partners, any sex (>75 vs. ≤75)**	**4.59**	**1.24–16.94**	**0.022**	3.05	0.54–17.16	0.21
**N. recent partners, any sex (>4 vs. ≤4)**	1.17	0.41–3.41	0.77			
**N. lifetime partners, receptive oral sex (>50 vs. ≤50)**	**3.73**	**1.13–12.25**	**0.03**	1.77	0.36–8.60	0.48
**N. recent partners, receptive oral sex (>3 vs. ≤3)**	0.83	0.28–2.45	0.74			
**Receptive anal sex, yes vs. no**	3.32	0.42–26.33	0.25			
**Condomless receptive anal sex, yes vs. no**	2.81	0.89–8.93	0.08			
**Receptive oral sex (ever), yes vs. no**	0.54	0.02–11.68	0.69			
**Receptive oral sex (recent), yes vs. no**	0.80	0.16–3.87	0.78			
**Condomless receptive oral sex, yes vs. no**	1.18	0.06–22.59	0.91			
**Ano-genital wart history, yes vs. no**	1.45	0.49–4.31	0.50			
**HIV status, positive vs. negative**	0.53	0.17–1.62	0.26			

COR, crude odds ratio; CI, confidence interval, AOR, adjusted odds ratio; COR, AOR and the respective *p* value are shown in bold when statistically significant.

## Data Availability

The data presented in this study are available on request from the corresponding author. The data are not publicly available due to the fact that they include personal data from vulnerable populations.

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
