# Peer review of "Concurrent and Concordant Anal and Oral Human PapillomaVirus Infections Are Not Associated with Sexual Behavior in At-Risk Males"

_pathogens, 2021, doi:10.3390/pathogens10101254_

Round 1
Reviewer 1 Report
Article by Rollo et al. (Manuscript ID: pathogens-1369077) examined prevalence of anal and oral HPV infection in men who have sex with men, a follow-up study to those previously published by the authors group trying to evaluate risk factors of HPV infections at various anatomical sites. About 30% of oropharyngeal carcinoma, a type of head and neck cancer, was found associated with oral HPV infection, and determining how HPV has acquired orally to those in at-risk group is much discussed in the HPV research field. Article in review dealt with survey of HPV prevalence in anal and oral sites in association with HIV infection, age, sex behavior and so on, attempting to determine routes that led to oral HPV infection. They showed that concurrent and concordant anal and oral HPV infection occurred in some subjects suggesting a link between anal and oral HPV infection. However, there were no factors that significantly associate with the prevalence of oral HPV except ages in concurrent infection. No sexual behaviors were found associated with the oral HPV infection led the authors to speculate the possibility that HPV was acquired by non-sexual route, auto-inoculation, which provided important aspects of HPV epidemiology. Although their research scope and results appeared scientifically sound, some part of the manuscript was difficult to read, which gave the impression that the authors’ conclusions were not properly addressed. This reviewer feels that the manuscript requires minor revision to polish English for publication in Pathogen.
Followings are noticeable confusing sentences that require authors’ attention.
Lines 52 and 53
“… it is crucial to investigate HPV infections simultaneously occurring at different anatomic sites.“ It is not clear from there description, and from the preceding sentences, that what was the point of investigating simultaneous, i.e. concurrent and concordant, infection of HPV. There must be better way to address it.
Lines 167 and 169
“Previously, we investigated anal and oral HPV infections in MSM populations, and compared HIV-infected vs. HIV-uninfected subjects in terms of prevalence at the two sites [14, 15]. Therefore, this aspect will not be further discussed here.”
The last sentence was confusing. It is not clear what “aspect” was not further discussed. Was it about “prevalence of two sites” or “HIV-infected versus uninfected”?
Lines 170-173
“In the present study, we found that prevalence of HPV infection by any HPV, high-risk types and HPV16 was significantly higher at anal than oral level, both among HIV-infected and HIV-uninfected MSM, with anal HPV exceeding oral HPV prevalence by 4 to 7 times.
The sentence was confusing and difficult to read. English needs to be polished.
Line 192-193
“The majority of our participants harbored non concurrent infections, with HPV being exclusively detected in the anal sample, except for two MSM. “
“ …except for two MSM”: Where is this data originated from, and what is the point of showing number (two) but not proportion (the majority), which make the sentence more confusing?

Author Response
Article by Rollo et al. (Manuscript ID: pathogens-1369077) examined prevalence of anal and oral HPV infection in men who have sex with men, a follow-up to the studies previously published by the authors group trying to evaluate risk factors of HPV infections at various anatomical sites. About 30% of oropharyngeal carcinoma, a type of head and neck cancer, was found associated with oral HPV infection, and determining how HPV has acquired orally to those in at-risk group is much discussed in the HPV research field. Article in review dealt with survey of HPV prevalence in anal and oral sites in association with HIV infection, age, sex behavior and so on, attempting to determine routes that led to oral HPV infection. They showed that concurrent and concordant anal and oral HPV infection occurred in some subjects suggesting a link between anal and oral HPV infection. However, there were no factors that significantly associate with the prevalence of oral HPV except ages in concurrent infection. No sexual behaviors were found associated with the oral HPV infection led the authors to speculate the possibility that HPV was acquired by non-sexual route, auto-inoculation, which provided important aspects of HPV epidemiology. Although their research scope and results appeared scientifically sound, some part of the manuscript was difficult to read, which gave the impression that the authors’ conclusions were not properly addressed. This reviewer feels that the manuscript requires minor revision to polish English for publication in Pathogen.
Authors’ response: We apologize if some parts of our manuscript were difficult to read. We have partially modified the conclusions, as follows “In conclusion, analyzing paired anal and oral samples of MSM, we observed that prevalence of anal HPV significantly exceeds prevalence of oral HPV, irrespective of the HIV status of the participants. Concurrent infections at anal and oral sites were found in around one-fifth of the study subjects. Age was the only predictor of concurrent infections. Genotype-specific concordance was infrequent but still observed.”
We have also modified the confusing sentences you pointed out.
Followings are noticeable confusing sentences that require authors’ attention.
Lines 52 and 53
“… it is crucial to investigate HPV infections simultaneously occurring at different anatomic sites.“ It is not clear from there description, and from the preceding sentences, that what was the point of investigating simultaneous, i.e. concurrent and concordant, infection of HPV. There must be better way to address it.
Authors’ response: We have now revised this part and modified it as follows: “We aimed to investigate whether or not HPV infections simultaneously occur at anal and oral sites in sexually active MSM, who habitually practice both anal and oral sex. This may contribute to clarify to which extent concurrent and concordant anal and oral HPV infections can be found among MSM. In addition, we evaluated the determinants of anal and oral HPV co-occurrence, also exploring possible associations with sexual behavior. This may explain whether or not sexual transmission plays a role in the occurrence of simultaneous anal and oral infections.”
Lines 167 and 169
“Previously, we investigated anal and oral HPV infections in MSM populations, and compared HIV-infected vs. HIV-uninfected subjects in terms of prevalence at the two sites [14, 15]. Therefore, this aspect will not be further discussed here.”
The last sentence was confusing. It is not clear what “aspect” was not further discussed. Was it about “prevalence of two sites” or “HIV-infected versus uninfected”?
Authors’ response: We have now revised this part and modified it as follows: “Previously, we investigated anal and oral HPV infections in MSM populations, and compared HIV-infected vs. HIV-uninfected subjects in terms of prevalence at the two sites. Therefore, the comparison between HIV-infected and HIV-uninfected MSM will not be further discussed here.”
Lines 170-173
“In the present study, we found that prevalence of HPV infection by any HPV, high-risk types and HPV16 was significantly higher at anal than oral level, both among HIV-infected and HIV-uninfected MSM, with anal HPV exceeding oral HPV prevalence by 4 to 7 times.
The sentence was confusing and difficult to read. English needs to be polished.
Authors’ response: We have now revised this part and modified it as follows: “In the present study, we found that prevalence of i) any type of HPV, ii) high-risk types and iii) HPV16 was significantly higher at anal than oral level, both among HIV-infected and HIV-uninfected MSM. In fact, anal HPV was found to exceed oral HPV prevalence by 4 to 7 times.”
Line 192-193
“The majority of our participants harbored non concurrent infections, with HPV being exclusively detected in the anal sample, except for two MSM. “
“ …except for two MSM”: Where is this data originated from, and what is the point of showing number (two) but not proportion (the majority), which make the sentence more confusing?
Authors’ response: We have now modified the Results in order to make it clear where these data originated from [“The majority of the participants (68.2% of the HIV-infected and 64.5% of the HIV-uninfected MSM) harbored non concurrent infections, i.e., they were HPV-positive exclusively at anal or oral level (Figure 2). However, the latter case was infrequent, being observed only in two out of 76 HIV-uninfected subjects (2.6%).]. In addition, we have also rephrased the sentence in the Discussion as follows: “The majority of our participants (approximately 65%) harbored non concurrent infections. In all the HIV-infected MSM with non concurrent infections, HPV was exclusively detected in the anal sample. Differently, 2.6% of the HIV-uninfected MSM harbored HPV only at oral level.”
Reviewer 2 Report
I have reviewed this manuscript and find it suitable for publication. I only have a few minor changes to suggest:
1) I would introduce the term for "IQR" (interquartile range) at or before line 63 before the abbreviation is introduced.
2) Get rid of the apostrophe at line(s) 175-76 by rewriting the line to something along the line of "the sexual behavior of the participants"
3) Get rid of the apostrophe s ("'s") on "manufacturer" at line 286.
4) At Table 1, right-align the "1" description.
5) Italicize the "et al" at lines 189, 203, and 207
6) Reference on line 308 is wrong format, convert "(Giuliani et al., 2021)"
to (ref) [23].
Author Response
I have reviewed this manuscript and find it suitable for publication. I only have a few minor changes to suggest:
Authors’ response: Thank you for finding our manuscript suitable for publication.
1) I would introduce the term for "IQR" (interquartile range) at or before line 63 before the abbreviation is introduced.
Authors’ response: the explanation for this abbreviation has been now introduced as requested.
2) Get rid of the apostrophe at line(s) 175-76 by rewriting the line to something along the line of "the sexual behavior of the participants"
Authors’ response: We have now replaced “participants’ sexual behavior” with “the sexual behavior of the participants” as suggested.
3) Get rid of the apostrophe s ("'s") on "manufacturer" at line 286.
Authors’ response: We have now replaced “manufacturer’s instructions” with “manufacturer instructions”, as suggested.
4) At Table 1, right-align the "1" description.
Authors’ response: since the format of our Table has been modified by the Editorial Office, we apologize but this request is not clear to us.
5) Italicize the "et al" at lines 189, 203, and 207
Authors’ response: this has been done, as suggested.
6) Reference on line 308 is wrong format, convert "(Giuliani et al., 2021)"
to (ref) [23].
Authors’ response: We have now replaced Giuliani et al., 2021 with [23], as suggested.